# More Effective Distributed ML via a Stale Synchronous Parallel Parameter Server

†**Qirong Ho**, †**James Cipar**, §**Henggang Cui**, †**Jin Kyu Kim**, †**Seunghak Lee**,
‡**Phillip B. Gibbons**, †**Garth A. Gibson**, §**Gregory R. Ganger**, †**Eric P. Xing**

| †School of Computer Science | §Electrical and Computer Engineering | ‡Intel Labs |
|---|---|---|
| Carnegie Mellon University | Carnegie Mellon University | Pittsburgh, PA 15213 |
| Pittsburgh, PA 15213 | Pittsburgh, PA 15213 | phillip.b.gibbons@intel.com |
| qho@, jcipar@, jinkyuk@, | hengganc@, ganger@ece.cmu.edu | |
| seunghak@, garth@, | | |
| epxing@cs.cmu.edu | | |

## Abstract

We propose a parameter server system for distributed ML, which follows a Stale Synchronous Parallel (SSP) model of computation that maximizes the time computational workers spend doing useful work on ML algorithms, while still providing correctness guarantees. The parameter server provides an easy-to-use shared interface for read/write access to an ML model's values (parameters and variables), and the SSP model allows distributed workers to read older, stale versions of these values from a local cache, instead of waiting to get them from a central storage. This significantly increases the proportion of time workers spend computing, as opposed to waiting. Furthermore, the SSP model ensures ML algorithm correctness by limiting the maximum age of the stale values. We provide a proof of correctness under SSP, as well as empirical results demonstrating that the SSP model achieves faster algorithm convergence on several different ML problems, compared to fully-synchronous and asynchronous schemes.

## 1   Introduction

Modern applications awaiting next generation machine intelligence systems have posed unprecedented scalability challenges. These scalability needs arise from at least two aspects: 1) massive data volume, such as societal-scale social graphs [10, 25] with up to hundreds of millions of nodes; and 2) massive model size, such as the Google Brain deep neural network [9] containing billions of parameters. Although there exist means and theories to support reductionist approaches like subsampling data or using small models, there is an imperative need for sound and effective distributed ML methodologies for users who cannot be well-served by such shortcuts. Recent efforts towards distributed ML have made significant advancements in two directions: (1) Leveraging existing common but simple distributed systems to implement parallel versions of a limited selection of ML models, that can be shown to have strong theoretical guarantees under parallelization schemes such as cyclic delay [17, 1], model pre-partitioning [12], lock-free updates [21], bulk synchronous parallel [5], or even no synchronization [28] — these schemes are simple to implement but may under-exploit the full computing power of a distributed cluster. (2) Building high-throughput distributed ML architectures or algorithm implementations that feature significant systems contributions but relatively less theoretical analysis, such as GraphLab [18], Spark [27], Pregel [19], and YahooLDA [2].

While the aforementioned works are significant contributions in their own right, a naturally desirable goal for distributed ML is to pursue a system that (1) can maximally unleash the combined computational power in a cluster of any given size (by spending more time doing useful computation and less time waiting for communication), (2) supports inference for a broad collection of ML methods, and (3) enjoys correctness guarantees. In this paper, we explore a path to such a system using the

idea of a *parameter server* [22, 2], which we define as the combination of a shared key-value store that provides a centralized storage model (which may be implemented in a distributed fashion) with a synchronization model for reading/updating model values. The key-value store provides easy-to-program read/write access to shared parameters needed by all workers, and the synchronization model maximizes the time each worker spends on useful computation (versus communication with the server) while still providing algorithm correctness guarantees.

Towards this end, we propose a parameter server using a *Stale Synchronous Parallel* (SSP) model of computation, for distributed ML algorithms that are parallelized into many computational workers (technically, threads) spread over many machines. In SSP, workers can make updates $\delta$ to a parameter[1] $\theta$, where the updates follow an associative, commutative form $\theta \leftarrow \theta + \delta$. Hence, the current true value of $\theta$ is just the sum over updates $\delta$ from all workers. When a worker asks for $\theta$, the SSP model will give it a *stale* (i.e. delayed) version of $\theta$ that excludes recent updates $\delta$. More formally, a worker reading $\theta$ at iteration $c$ will see the effects of all $\delta$ from iteration 0 to $c - s - 1$, where $s \geq 0$ is a user-controlled staleness threshold. In addition, the worker may get to see some recent updates beyond iteration $c - s - 1$. The idea is that SSP systems should deliver as many updates as possible, without missing any updates older than a given age — a concept referred to as *bounded staleness* [24]. The practical effect of this is twofold: (1) workers can perform more computation instead of waiting for other workers to finish, and (2) workers spend less time communicating with the parameter server, and more time doing useful computation. Bounded staleness distinguishes SSP from cyclic-delay systems [17, 1] (where $\theta$ is read with inflexible staleness), Bulk Synchronous Parallel (BSP) systems like Hadoop (workers must wait for each other at the end of every iteration), or completely asynchronous systems [2] (workers never wait, but $\theta$ has no staleness guarantees).

We implement an SSP parameter server with a table-based interface, called SSPtable, that supports a wide range of distributed ML algorithms for many models and applications. SSPtable itself can also be run in a distributed fashion, in order to (a) increase performance, or (b) support applications where the parameters $\theta$ are too large to fit on one machine. Moreover, SSPtable takes advantage of bounded staleness to maximize ML algorithm performance, by reading the parameters $\theta$ from caches on the worker machines whenever possible, and only reading $\theta$ from the parameter server when the SSP model requires it. Thus, workers (1) spend less time waiting for each other, and (2) spend less time communicating with the parameter server. Furthermore, we show that SSPtable (3) helps slow, straggling workers to catch up, providing a systems-based solution to the "last reducer" problem on systems like Hadoop (while we note that theory-based solutions are also possible). SSPtable can be run on multiple server machines (called "shards"), thus dividing its workload over the cluster; in this manner, SSPtable can (4) service more workers simultaneously, and (5) support very large models that cannot fit on a single machine. Finally, the SSPtable server program can also be run on worker machines, which (6) provides a simple but effective strategy for allocating machines between workers and the parameter server.

Our theoretical analysis shows that (1) SSP generalizes the bulk synchronous parallel (BSP) model, and that (2) stochastic gradient algorithms (e.g. for matrix factorization or topic models) under SSP not only converge, but do so at least as fast as cyclic-delay systems [17, 1] (and potentially even faster depending on implementation). Furthermore, our implementation of SSP, SSPtable, supports a wide variety of algortihms and models, and we demonstrate it on several popular ones: (a) Matrix Factorization with stochastic gradient descent [12], (b) Topic Modeling with collapsed Gibbs sampling [2], and (c) Lasso regression with parallelized coordinate descent [5]. Our experimental results show that, for these 3 models and algorithms, (i) SSP yields faster convergence than BSP (up to several times faster), and (ii) SSP yields faster convergence than a fully asynchronous (i.e. no staleness guarantee) system. We explain SSPtable's better performance in terms of algorithm progress per iteration (quality) and iterations executed per unit time (quantity), and show that SSPtable hits a "sweet spot" between quality and quantity that is missed by BSP and fully asynchronous systems.

## 2   Stale Synchronous Parallel Model of Computation

We begin with an informal explanation of SSP: assume a collection of $P$ workers, each of which makes additive updates to a shared parameter $\mathbf{x} \leftarrow \mathbf{x} + \mathbf{u}$ at regular intervals called *clocks*. Clocks are similar to iterations, and represent some unit of progress by an ML algorithm. Every worker

has its own integer-valued clock $c$, and workers only commit their updates at the end of each clock. Updates may not be immediately visible to other workers trying to read $\mathbf{x}$ — in other words, workers only see effects from a "stale" subset of updates. The idea is that, with staleness, workers can retrieve updates from caches on the same machine (fast) instead of querying the parameter server over the network (slow). Given a user-chosen staleness threshold $s \geq 0$, SSP enforces the following *bounded staleness* conditions (see Figure 1 for a graphical illustration):

- The slowest and fastest workers must be $\leq s$ clocks apart — otherwise, the fastest worker is forced to wait for the slowest worker to catch up.
- When a worker with clock $c$ commits an update $\mathbf{u}$, that $\mathbf{u}$ is timestamped with time $c$.
- When a worker with clock $c$ reads $\mathbf{x}$, it will always see effects from all $\mathbf{u}$ with timestamp $\leq c - s - 1$. It may also see some $\mathbf{u}$ with timestamp $> c - s - 1$ from other workers.
- Read-my-writes: A worker $p$ will always see the effects of its own updates $\mathbf{u}_p$.

Since the fastest and slowest workers are $\leq s$ clocks apart, a worker reading $\mathbf{x}$ at clock $c$ will see all updates with timestamps in $[0, c - s - 1]$, plus a (possibly empty) "adaptive" subset of updates in the range $[c - s, c + s - 1]$. Note that when $s = 0$, the "guaranteed" range becomes $[0, c - 1]$ while the adaptive range becomes empty, which is exactly the Bulk Synchronous Parallel model of computation. Let us look at how SSP applies to an example ML algorithm.

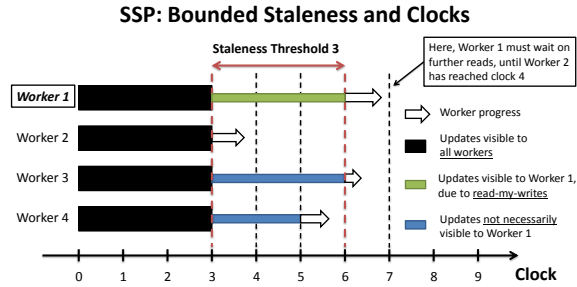

Figure 1: Bounded Staleness under the SSP Model

## 2.1 An example: Stochastic Gradient Descent for Matrix Problems

The Stochastic Gradient Descent (SGD) [17, 12] algorithm optimizes an objective function by applying gradient descent to random subsets of the data. Consider a matrix completion task, which involves decomposing an $N \times M$ matrix $D$ into two low-rank matrices $LR \approx D$, where $L, R$ have sizes $N \times K$ and $K \times M$ (for a user-specified $K$). The data matrix $D$ may have missing entries, corresponding to missing data. Concretely, $D$ could be a matrix of users against products, with $D_{ij}$ representing user $i$'s rating of product $j$. Because users do not rate all possible products, the goal is to predict ratings for missing entries $D_{ab}$ given known entries $D_{ij}$. If we found low-rank matrices $L, R$ such that $L_i. \cdot R_{\cdot j} \approx D_{ij}$ for all known entries $D_{ij}$, we could then predict $D_{ab} = L_a. \cdot R_{\cdot b}$ for unknown entries $D_{ab}$.

To perform the decomposition, let us minimize the squared difference between each known entry $D_{ij}$ and its prediction $L_i. \cdot R_{\cdot j}$ (note that other loss functions and regularizers are also possible):

$$\min_{L,R} \sum_{(i,j)\in\text{Data}} \left\| D_{ij} - \sum_{k=1}^{K} L_{ik} R_{kj} \right\|^2 . \tag{1}$$

As a first step towards SGD, consider solving Eq (1) using coordinate gradient descent on $L, R$:

$$\frac{\partial \mathbf{O}_{\text{MF}}}{\partial L_{ik}} = \sum_{(a,b)\in\text{Data}} \delta(a = i) \left[ -2D_{ab}R_{kb} + 2L_a. \cdot R_{\cdot b}R_{kb} \right], \qquad \frac{\partial \mathbf{O}_{\text{MF}}}{\partial R_{kj}} = \sum_{(a,b)\in\text{Data}} \delta(b = j) \left[ -2D_{ab}L_{ak} + 2L_a. \cdot R_{\cdot b}L_{ak} \right]$$

where $\mathbf{O}_{\text{MF}}$ is the objective in Eq(1), and $\delta(a = i)$ equals 1 if $a = i$, and 0 otherwise. This can be transformed into an SGD algorithm by replacing the full sum over entries $(a, b)$ with a subsample (with appropriate reweighting). The entries $D_{ab}$ can then be distributed over multiple workers, and their gradients computed in parallel [12].

We assume that $D$ is "tall", i.e. $N > M$ (or transpose $D$ so this is true), and partition the rows of $D$ and $L$ over the processors. Only $R$ needs to be shared among all processors, so we let it be the SSP shared parameter $\mathbf{x} := R$. SSP allows many workers to read/write to $R$ with minimal waiting, though the workers will only see stale values of $R$. This tradeoff is beneficial because without staleness, the workers must wait for a long time when reading $R$ from the server (as our experiments will show). While having stale values of $R$ decreases convergence progress per iteration, SSP more than makes up by enabling significantly more iterations per minute, compared to fully synchronous systems. Thus, SSP yields more convergence progress per minute, i.e. faster convergence.

Note that SSP is not limited to stochastic gradient matrix algorithms: it can also be applied to parallel collapsed sampling on topic models [2] (by storing the word-topic and document-topic tables in $\mathbf{x}$), parallel coordinate descent on Lasso regression [5] (by storing the regression coefficients $\beta$ in $\mathbf{x}$), as well as any other parallel algorithm or model with shared parameters that all workers need read/write access to. Our experiments will show that SSP performs better than bulk synchronous parallel and asynchronous systems for matrix completion, topic modeling and Lasso regression.

## 3   SSPtable: an Efficient SSP System

An ideal SSP implementation would fully exploit the lee-way granted by the SSP's bounded staleness property, in order to balance the time workers spend waiting on reads with the need for freshness in the shared data. This section describes our initial implementation of SSPtable, which is a parameter server conforming to the SSP model, and that can be run on many server machines at once (distributed). Our experiments with this SSPtable implementation shows that SSP can indeed improve convergence rates for several ML models and algorithms, while further tuning of cache management policies could further improve the performance of SSPtable.

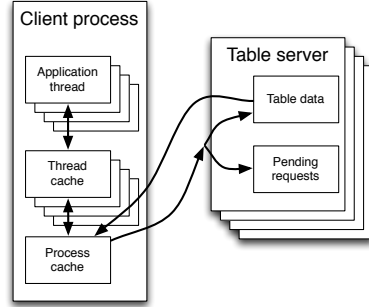

Figure 2: Cache structure of SSPtable, with multiple server shards

SSPtable follows a distributed client-server architecture. Clients access shared parameters using a client library, which maintains a machine-wide *process cache* and optional per-thread[2] thread caches (Figure 2); the latter are useful for improving performance, by reducing inter-thread synchronization (which forces workers to wait) when a client ML program executes multiple worker threads on each of multiple cores of a client machine. The server parameter state is divided (sharded) over multiple server machines, and a normal configuration would include a server process on each of the client machines. Programming with SSPtable follows a simple table-based API for reading/writing to shared parameters $\mathbf{x}$ (for example, the matrix $R$ in the SGD example of Section 2.1):

- **Table Organization:** SSPtable supports an unlimited number of *tables*, which are divided into *rows*, which are further subdivided into *elements*. These tables are used to store $\mathbf{x}$.
- `read_row(table,row,s)`: Retrieve a table-row with staleness threshold `s`. The user can then query individual row elements.
- `inc(table,row,el,val)`: Increase a table-row-element by `val`, which can be negative. These changes are not propagated to the servers until the next call to `clock()`.
- `clock()`: Inform all servers that the current thread/processor has completed one clock, and commit all outstanding `inc()`s to the servers.

Any number of `read_row()` and `inc()` calls can be made in-between calls to `clock()`. Different thread workers are permitted to be at different clocks, however, bounded staleness requires that the fastest and slowest threads be no more than `s` clocks apart. In this situation, SSPtable forces the fastest thread to block (i.e. wait) on calls to `read_row()`, until the slowest thread has caught up. To maintain the "read-my-writes" property, we use a write-back policy: all writes are immediately committed to the thread caches, and are flushed to the process cache and servers upon `clock()`.

To maintain bounded staleness while minimizing wait times on `read_row()` operations, SSPtable uses the following cache protocol: Let every table-row in a thread or process cache be endowed with a clock $r_{thread}$ or $r_{proc}$ respectively. Let every thread worker be endowed with a clock $c$, equal to the number of times it has called `clock()`. Finally, define the server clock $c_{server}$ to be the minimum over all thread clocks $c$. When a thread with clock $c$ requests a table-row, it first checks its thread cache. If the row is cached with clock $r_{thread} \geq c - s$, then it reads the row. Otherwise, it checks the process cache next — if the row is cached with clock $r_{proc} \geq c - s$, then it reads the row. At this point, no network traffic has been incurred yet. However, if both caches miss, then a network request is sent to the server (which forces the thread to wait for a reply). The server returns its view of the table-row as well as the clock $c_{server}$. Because the fastest and slowest threads can be no more than $s$ clocks apart, and because a thread's updates are sent to the server whenever it calls `clock()`, the returned server view always satisfies the bounded staleness requirements for the

asking thread. After fetching a row from the server, the corresponding entry in the thread/process caches and the clocks $r_{thread}, r_{proc}$ are then overwritten with the server view and clock $c_{server}$.

A beneficial consequence of this cache protocol is that the slowest thread only performs costly server reads every $s$ clocks. Faster threads may perform server reads more frequently, and as frequently as every clock if they are consistently waiting for the slowest thread's updates. This distinction in work per thread does not occur in BSP, wherein every thread must read from the server on every clock. Thus, SSP not only reduces overall network traffic (thus reducing wait times for all server reads), but also allows slow, straggler threads to avoid server reads in some iterations. Hence, the slow threads naturally catch up — in turn allowing fast threads to proceed instead of waiting for them. In this manner, SSP maximizes the time each machine spends on useful computation, rather than waiting.

## 4  Theoretical Analysis of SSP

Formally, the SSP model supports operations $\mathbf{x} \leftarrow \mathbf{x} \oplus (z \cdot \mathbf{y})$, where $\mathbf{x}, \mathbf{y}$ are members of a ring with an abelian operator $\oplus$ (such as addition), and a multiplication operator $\cdot$ such that $z \cdot \mathbf{y} = \mathbf{y}'$ where $\mathbf{y}'$ is also in the ring. In the context of ML, we shall focus on addition and multiplication over real vectors $\mathbf{x}, \mathbf{y}$ and scalar coefficients $z$, i.e. $\mathbf{x} \leftarrow \mathbf{x} + (z\mathbf{y})$; such operations can be found in the update equations of many ML inference algorithms, such as gradient descent [12], coordinate descent [5] and collapsed Gibbs sampling [2]. In what follows, we shall informally refer to $\mathbf{x}$ as the "system state", $\mathbf{u} = z\mathbf{y}$ as an "update", and to the operation $\mathbf{x} \leftarrow \mathbf{x} + \mathbf{u}$ as "writing an update".

We assume that $P$ workers write updates at regular time intervals (referred to as "clocks"). Let $\mathbf{u}_{p,c}$ be the update written by worker $p$ at clock $c$ through the write operation $\mathbf{x} \leftarrow \mathbf{x} + \mathbf{u}_{p,c}$. The updates $\mathbf{u}_{p,c}$ are a function of the system state $\mathbf{x}$, and under the SSP model, different workers will "see" different, noisy versions of the true state $\mathbf{x}$. Let $\tilde{\mathbf{x}}_{p,c}$ be the noisy state read by worker $p$ at clock $c$, implying that $\mathbf{u}_{p,c} = G(\tilde{\mathbf{x}}_{p,c})$ for some function $G$. We now formally re-state *bounded staleness*, which is the key SSP condition that bounds the possible values $\tilde{\mathbf{x}}_{p,c}$ can take:

**SSP Condition (Bounded Staleness):**  Fix a staleness $s$. Then, the noisy state $\tilde{\mathbf{x}}_{p,c}$ is equal to

$$\tilde{\mathbf{x}}_{p,c} = \mathbf{x}_0 + \underbrace{\left[ \sum_{c'=1}^{c-s-1} \sum_{p'=1}^{P} \mathbf{u}_{p',c'} \right]}_{\text{guaranteed pre-window updates}} + \underbrace{\left[ \sum_{c'=c-s}^{c-1} \mathbf{u}_{p,c'} \right]}_{\text{guaranteed read-my-writes updates}} + \underbrace{\left[ \sum_{(p',c') \in \mathcal{S}_{p,c}} \mathbf{u}_{p',c'} \right]}_{\text{best-effort in-window updates}}, \qquad (2)$$

where $\mathcal{S}_{p,c} \subseteq \mathcal{W}_{p,c} = ([1, P] \setminus \{p\}) \times [c - s, c + s - 1]$ is some subset of the updates $\mathbf{u}$ written in the width-$2s$ "window" $\mathcal{W}_{p,c}$, which ranges from clock $c - s$ to $c + s - 1$ and does not include updates from worker $p$. In other words, the noisy state $\tilde{\mathbf{x}}_{p,c}$ consists of three parts:

1. Guaranteed "pre-window" updates from clock 0 to $c - s - 1$, over all workers.
2. Guaranteed "read-my-writes" set $\{(p, c - s), \ldots, (p, c - 1)\}$ that covers all "in-window" updates made by the querying worker[3] $p$.
3. Best-effort "in-window" updates $\mathcal{S}_{p,c}$ from the width-$2s$ window[4] $[c - s, c + s - 1]$ (not counting updates from worker $p$). An SSP implementation should try to deliver as many updates from $\mathcal{S}_{p,c}$ as possible, but may choose not to depending on conditions.

Notice that $\mathcal{S}_{p,c}$ is specific to worker $p$ at clock $c$; other workers at different clocks will observe different $\mathcal{S}$. Also, observe that SSP generalizes the Bulk Synchronous Parallel (BSP) model:

**BSP Corollary:**  Under zero staleness $s = 0$, SSP reduces to BSP. **Proof:** $s = 0$ implies $[c, c + s - 1] = \emptyset$, and therefore $\tilde{\mathbf{x}}_{p,c}$ exactly consists of all updates until clock $c - 1$. $\square$

Our key tool for convergence analysis is to define a reference sequence of states $\mathbf{x}_t$, informally referred to as the "true" sequence (this is different and unrelated to the SSPtable server's view):

$$\mathbf{x}_t = \mathbf{x}_0 + \sum_{t'=0}^{t} \mathbf{u}_{t'}, \qquad \text{where} \quad \mathbf{u}_t := \mathbf{u}_{t \bmod P, \lfloor t/P \rfloor}.$$

In other words, we sum updates by first looping over workers ($t \bmod P$), then over clocks $\lfloor t/P \rfloor$. We can now bound the difference between the "true" sequence $\mathbf{x}_t$ and the noisy views $\tilde{\mathbf{x}}_{p,c}$:

**Lemma 1:** Assume $s \geq 1$, and let $\tilde{\mathbf{x}}_t := \tilde{\mathbf{x}}_{t \bmod P, \lfloor t/P \rfloor}$, so that

$$\tilde{\mathbf{x}}_t = \mathbf{x}_t - \underbrace{\left[\sum_{i \in \mathcal{A}_t} \mathbf{u}_i\right]}_{\text{missing updates}} + \underbrace{\left[\sum_{i \in \mathcal{B}_t} \mathbf{u}_i\right]}_{\text{extra updates}}, \tag{3}$$

where we have decomposed the difference between $\tilde{\mathbf{x}}_t$ and $\mathbf{x}_t$ into $\mathcal{A}_t$, the index set of updates $\mathbf{u}_i$ that are missing from $\tilde{\mathbf{x}}_t$ (w.r.t. $\mathbf{x}_t$), and $\mathcal{B}_t$, the index set of "extra" updates in $\tilde{\mathbf{x}}_t$ but not in $\mathbf{x}_t$. We then claim that $|\mathcal{A}_t| + |\mathcal{B}_t| \leq 2s(P-1)$, and furthermore, $\min(\mathcal{A}_t \cup \mathcal{B}_t) \geq \max(1, t - (s+1)P)$, and $\max(\mathcal{A}_t \cup \mathcal{B}_t) \leq t + sP$.

**Proof:** Comparing Eq. (3) with (2), we see that the extra updates obey $\mathcal{B}_t \subseteq \mathcal{S}_{t \bmod P, \lfloor t/P \rfloor}$, while the missing updates obey $\mathcal{A}_t \subseteq (\mathcal{W}_{t \bmod P, \lfloor t/P \rfloor} \setminus \mathcal{S}_{t \bmod P, \lfloor t/P \rfloor})$. Because $|\mathcal{W}_{t \bmod P, \lfloor t/P \rfloor}| = 2s(P-1)$, the first claim immediately follows. The second and third claims follow from looking at the left- and right-most boundaries of $\mathcal{W}_{t \bmod P, \lfloor t/P \rfloor}$. $\square$

Lemma 1 basically says that the "true" state $\mathbf{x}_t$ and the noisy state $\tilde{\mathbf{x}}_t$ only differ by at most $2s(P-1)$ updates $\mathbf{u}_t$, and that these updates cannot be more than $(s+1)P$ steps away from $t$. These properties can be used to prove convergence bounds for various algorithms; in this paper, we shall focus on stochastic gradient descent SGD [17]:

**Theorem 1 (SGD under SSP):** Suppose we want to find the minimizer $\mathbf{x}^*$ of a convex function $f(\mathbf{x}) = \frac{1}{T}\sum_{t=1}^{T} f_t(\mathbf{x})$, via gradient descent on one component $\nabla f_t$ at a time. We assume the components $f_t$ are also convex. Let $\mathbf{u}_t := -\eta_t \nabla f_t(\tilde{\mathbf{x}}_t)$, where $\eta_t = \frac{\sigma}{\sqrt{t}}$ with $\sigma = \frac{F}{L\sqrt{2(s+1)P}}$ for certain constants $F, L$. Then, under suitable conditions ($f_t$ are $L$-Lipschitz and the distance between two points $D(x\|x') \leq F^2$),

$$R[\mathbf{X}] := \left[\frac{1}{T}\sum_{t=1}^{T} f_t(\tilde{\mathbf{x}}_t)\right] - f(\mathbf{x}^*) \leq 4FL\sqrt{\frac{2(s+1)P}{T}}$$

This means that the noisy worker views $\tilde{\mathbf{x}}_t$ converge in expectation to the true view $\mathbf{x}^*$ (as measured by the function $f()$, and at rate $\mathcal{O}(T^{-1/2})$). We defer the proof to the appendix, noting that it generally follows the analysis in Langford *et al.* [17], except in places where Lemma 1 is involved. Our bound is also similar to [17], except that (1) their fixed delay $\tau$ has been replaced by our staleness upper bound $2(s+1)P$, and (2) we have shown convergence of the noisy worker views $\tilde{\mathbf{x}}_t$ rather than a true sequence $\mathbf{x}_t$. Furthermore, because the constant factor $2(s+1)P$ is only an upper bound to the number of erroneous updates, SSP's rate of convergence has a potentially tighter constant factor than Langford *et al.*'s fixed staleness system (details are in the appendix).

## 5 Experiments

We show that the SSP model outperforms fully-synchronous models such as Bulk Synchronous Parallel (BSP) that require workers to wait for each other on every iteration, as well as asynchronous models with no model staleness guarantees. The general experimental details are:

- **Computational models and implementation:** SSP, BSP and Asynchronous[5]. We used SSPtable for the first two (BSP is just staleness 0 under SSP), and implemented the Asynchronous model using many of the caching features of SSPtable (to keep the implementations comparable).
- **ML models (and parallel algorithms):** LDA Topic Modeling (collapsed Gibbs sampling), Matrix Factorization (stochastic gradient descent) and Lasso regression (coordinate gradient descent). All algorithms were implemented using SSPtable's parameter server interface. For TM and MF, we ran the algorithms in a "full batch" mode (where the algorithm's workers collectively touch every data point once per `clock()`), as well as a "10% minibatch" model (workers touch 10% of the data per `clock()`). Due to implementation limitations, we did not run Lasso under the Async model.
- **Datasets:** Topic Modeling: New York Times ($N = 100m$ tokens, $V = 100k$ terms, $K = 100$ topics), Matrix Factorization: NetFlix (480k-by-18k matrix with 100m nonzeros, rank $K = 100$ decomposition), Lasso regression: Synthetic dataset ($N = 500$ samples with $P = 400k$ features[6]). We use a static data partitioning strategy explained in the Appendix.
- **Compute cluster:** Multi-core blade servers connected by 10 Gbps Ethernet, running VMware ESX. We use one virtual machine (VM) per physical machine. Each VM is configured with 8 cores (either 2.3GHz or 2.5GHz each) and 23GB of RAM, running on top of Debian Linux 7.0.

**Convergence Speed.** Figure 3 shows objective vs. time plots for the three ML algorithms, over several machine configurations. We are interested in how long each algorithm takes to reach a given objective value, which corresponds to drawing horizontal lines on the plots. On each plot, we show curves for BSP (zero staleness), Async, and SSP for the best staleness value $\geq 1$ (we generally omit the other SSP curves to reduce clutter). In all cases except Topic Modeling with 8 VMs, SSP converges to a given objective value faster than BSP or Async. The gap between SSP and the other systems increases with more VMs and smaller data batches, because both of these factors lead to increased network communication — which SSP is able to reduce via staleness. We also provide a scalability-with-$N$-machines plot in the Appendix.

**Computation Time vs Network Waiting Time.** To understand why SSP performs better, we look at how the Topic Modeling (TM) algorithm spends its time during a fixed number of `clock()`s. In the 2nd row of Figure 3, we see that for any machine configuration, the TM algorithm spends roughly the same amount of time on useful computation, regardless of the staleness value. However, the time spent waiting for network communication drops rapidly with even a small increase in staleness, allowing SSP to execute `clock()`s more quickly than BSP (staleness 0). Furthermore, the ratio of network-to-compute time increases as we add more VMs, or use smaller data batches. At 32 VMs and $10\%$ data minibatches, the TM algorithm under BSP spends *six times* more time on network communications than computation. In contrast, the optimal value of staleness, 32, exhibits a 1:1 ratio of communication to computation. Hence, the value of SSP lies in allowing ML algorithms to perform far more useful computations per second, compared to the BSP model (e.g. Hadoop). Similar observations hold for the MF and Lasso applications (graphs not shown for space reasons).

**Iteration Quantity and Quality.** The network-compute ratio only partially explains SSP's behavior; we need to examine each `clock()`'s behavior to get a full picture. In the 3rd row of Figure 3, we plot the number of clocks executed per worker per unit time for the TM algorithm, as well as the objective value at each clock. Higher staleness values increase the number of clocks executed per unit time, but decrease each clock's progress towards convergence (as suggested by our theory); MF and Lasso also exhibit similar behavior (graphs not shown). Thus, staleness is a tradeoff between iteration quantity and quality — and because the iteration rate exhibits diminishing returns with higher staleness values, there comes a point where additional staleness starts to hurt the rate of convergence per time. This explains why the best staleness value in a given setting is some constant $0 < s < \infty$ — hence, SSP can hit a "sweet spot" between quality/quantity that BSP and Async do not achieve. Automatically finding this sweet spot for a given problem is a subject for future work.

## 6   Related Work and Discussion

The idea of staleness has been explored before: in ML academia, it has been analyzed in the context of cyclic-delay architectures [17, 1], in which machines communicate with a central server (or each other) under a fixed schedule (and hence fixed staleness). Even the bulk synchronous parallel (BSP) model inherently produces stale communications, the effects of which have been studied for algorithms such as Lasso regression [5] and topic modeling [2]. Our work differs in that SSP advocates *bounded* (rather than fixed) staleness to allow higher computational throughput via local machine caches. Furthermore, SSP's performance does not degrade when parameter updates frequently collide on the same vector elements, unlike asynchronous lock-free systems [21]. We note that staleness has been informally explored in the industrial setting at large scales; our work provides a first attempt at rigorously justifying staleness as a sound ML technique.

Distributed platforms such as Hadoop and GraphLab [18] are popular for large-scale ML. The biggest difference between them and SSPtable is the programming model — Hadoop uses a stateless map-reduce model, while GraphLab uses stateful vertex programs organized into a graph. In contrast, SSPtable provides a convenient shared-memory programming model based on a table/matrix API, making it easy to convert single-machine parallel ML algorithms into distributed versions. In particular, the algorithms used in our experiments — LDA, MF, Lasso — are all straightforward conversions of single-machine algorithms. Hadoop's BSP execution model is a special case of SSP, making SSPtable more general in that regard; however, Hadoop also provides fault-tolerance and distributed filesystem features that SSPtable does not cover. Finally, there exist special-purpose tools such as Vowpal Wabbit [16] and YahooLDA [2]. Whereas these systems have been targeted at a subset of ML algorithms, SSPtable can be used by any ML algorithm that tolerates stale updates.

The distributed systems community has typically examined staleness in the context of consistency models. The TACT model [26] describes consistency along three dimensions: numerical error, order error, and staleness. Other work [24] attempts to classify existing systems according to a number

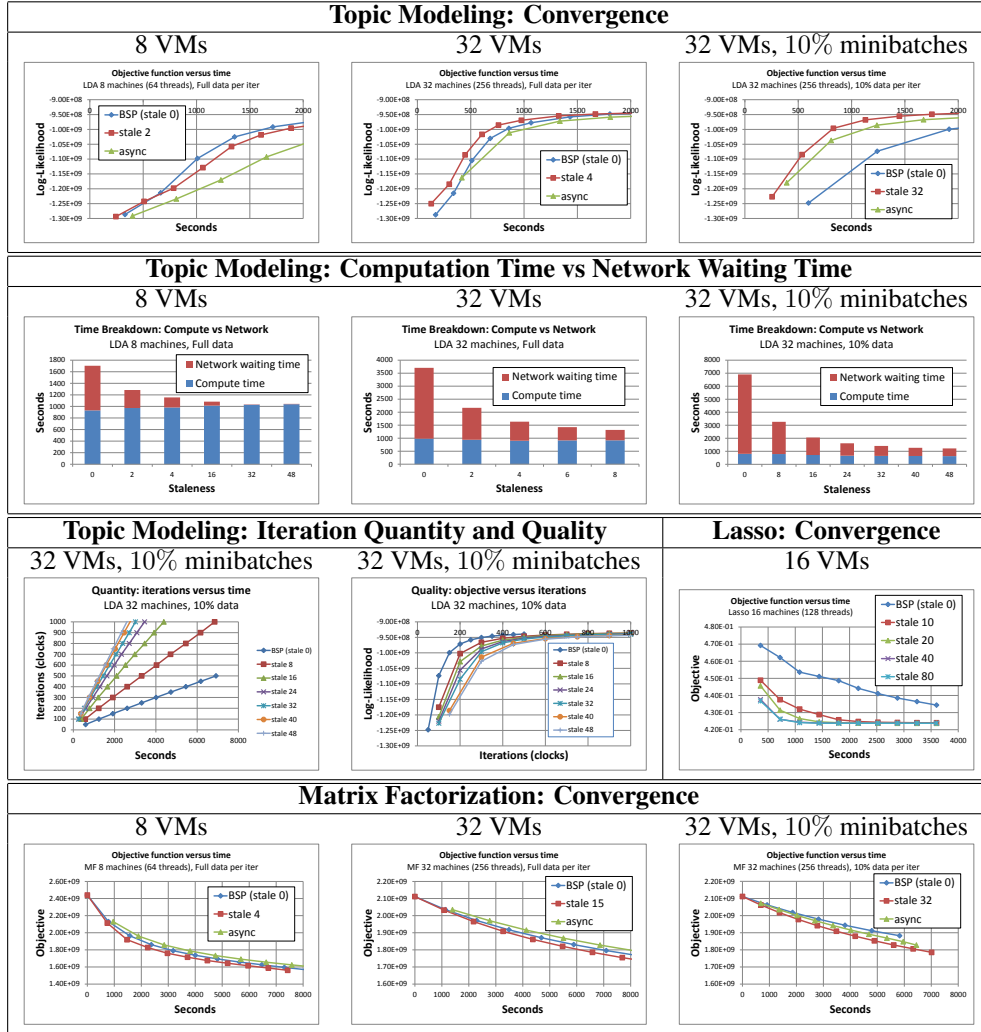

Figure 3: **Experimental results:** SSP, BSP and Asynchronous parameter servers running Topic Modeling, Matrix Factorization and Lasso regression. The *Convergence* graphs plot objective function (i.e. solution quality) against time. For Topic Modeling, we also plot computation time vs network waiting time, as well as how staleness affects iteration (clock) frequency (*Quantity*) and objective improvement per iteration (*Quality*).

of consistency properties, specifically naming the concept of bounded staleness. The vector clocks used in SSPtable are similar to those in Fidge [11] and Mattern [20], which were in turn inspired by Lamport clocks [15]. However, SSPtable uses vector clocks to track the freshness of the data, rather than causal relationships between updates. [8] gives an informal definition of the SSP model, motivated by the need to reduce straggler effects in large compute clusters.

In databases, bounded staleness has been applied to improve update and query performance. Lazy-Base [7] allows staleness bounds to be configured on a per-query basis, and uses this relaxed staleness to improve both query and update performance. FAS [23] keeps data replicated in a number of databases, each providing a different freshness/performance tradeoff. Data stream warehouses [13] collect data about timestamped events, and provide different consistency depending on the freshness of the data. Staleness (or freshness/timeliness) has also been applied in other fields such as sensor networks [14], dynamic web content generation [3], web caching [6], and information systems [4].

**Acknowledgments**

Qirong Ho is supported by an NSS-PhD Fellowship from A-STAR, Singapore. This work is supported in part by NIH 1R01GM087694 and 1R01GM093156, DARPA FA87501220324, and NSF IIS1111142 to Eric P. Xing. We thank the member companies of the PDL Consortium (Actifio, APC, EMC, Emulex, Facebook, Fusion-IO, Google, HP, Hitachi, Huawei, Intel, Microsoft, NEC, NetApp, Oracle, Panasas, Samsung, Seagate, Symantec, VMware, Western Digital) for their interest, insights, feedback, and support. This work is supported in part by Intel via the Intel Science and Technology Center for Cloud Computing (ISTC-CC) and hardware donations from Intel and NetApp.

## Footnotes

[1] For example, the parameter $\theta$ might be the topic-word distributions in LDA, or the factor matrices in a matrix decomposition, while the updates $\delta$ could be adding or removing counts to topic-word or document-word tables in LDA, or stochastic gradient steps in a matrix decomposition.

[2] We assume that every computation thread corresponds to one ML algorithm worker.

[3] This is a "read-my-writes" or self-synchronization property, i.e. workers will always see any updates they make. Having such a property makes sense because self-synchronization does not incur a network cost.

[4] The width $2s$ is only an upper bound for the slowest worker. The fastest worker with clock $c_{max}$ has a width-$s$ window $[c_{max} - s, c_{max} - 1]$, simply because no updates for clocks $\geq c_{max}$ have been written yet.

[5] The Asynchronous model is used in many ML frameworks, such as YahooLDA [2] and HogWild! [21].

[6] This is the largest data size we could get the Lasso algorithm to converge on, under ideal BSP conditions.

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
