[Supplementary Material]

# Appendix
# More Effective Distributed ML via a Stale Synchronous Parallel Parameter Server

## A  Additional Experimental Details

### A.1  Data Partitioning

For all three ML algorithms (topic modeling, matrix factorization and Lasso regression), we partition data statically and equally over worker threads. In other words, if there are $N$ data points and $P$ threads, then every thread gets $N/P$ data points. For topic modeling, a data point is one document. For matrix factorization, a data point is one row of the input matrix. Lasso is an exception — since it is a coordinate-parallel algorithm, we partition over input dimensions (columns) rather than data samples (rows). Note that SSP does not require static partitioning; dynamic strategies for load balancing are possible, and will likely improve algorithm performance further. We use static partitioning only to limit the number of experimental factors.

### A.2  Scalability with $N$ machines

Figure 1 shows how SSP scales with the number of machines used, on the topic modeling problem with a fixed dataset (NYtimes) and staleness (10). By using more machines, the algorithm reaches a given log-likelihood value (i.e. solution quality) more quickly; this can be seen by looking at the horizontal gridlines. For the specific settings mentioned, using 32 machines causes the algorithm to reach a given log-likelihood about 10 times as quickly as a single machine.

Figure 1: SSP scalability plot

# B  Proof of Theorem 1

We use slightly different definitions from the main paper, specifically:

$$f(x) := \sum_{t=1}^{T} f_t(x)$$

$$R[X] := \sum_{t=1}^{T} f_t(\tilde{x}_t) - f(x^*).$$

We also define $D\left(x \| x'\right) := \frac{1}{2}\|x - x'\|^2$. Compared to the main paper, $f(x)$ and $R[X]$ have no $1/T$ factor (this does not materially affect our results). We now re-state Theorem 1:

**Theorem 1 (SGD under SSP):**  Suppose we want to find the minimizer $\mathbf{x}^*$ of a convex function $f(\mathbf{x}) = \sum_{t=1}^{T} f_t(\mathbf{x})$, via gradient descent on one component $\nabla f_t$ at a time. We assume the components $f_t$ are also convex. Let $\mathbf{u}_t := -\eta_t \nabla f_t(\tilde{\mathbf{x}}_t)$, where $\eta_t = \frac{\sigma}{\sqrt{t}}$ with $\sigma = \frac{F}{L\sqrt{2(s+1)P}}$ for certain constants $F, L$. Then, assuming that $\|\nabla f_t(x)\| \leq L$ for all $t$ (i.e. $f_t$ are $L$-Lipschitz), and that $\max_{x,x' \in X} D\left(x \| x'\right) \leq F^2$ (the optimization problem has bounded diameter), we claim that

$$R[X] := \sum_{t=1}^{T} f_t(\tilde{x}_t) - f(x^*) \leq 4FL\sqrt{2(s+1)PT}$$

**Proof:**  The analysis follows Langford *et al.* (2009), except where the Lemma 1 from the main paper is involved. First,

$$
\begin{aligned}
R[X] &:= \sum_{t=1}^{T} f_t(\tilde{x}_t) - f_t(x^*) \\
&\leq \sum_{t=1}^{T} \langle \nabla f_t(\tilde{x}_t), \tilde{x}_t - x^* \rangle \qquad (f_t \text{ are convex}) \\
&= \sum_{t=1}^{T} \langle \tilde{g}_t, \tilde{x}_t - x^* \rangle.
\end{aligned}
$$

where we have defined $\tilde{g}_t := \nabla f_t(\tilde{x}_t)$. The high-level idea is to show that $R[X] \leq o(T)$, which implies $\mathbb{E}_t\left[f_t(\tilde{x}_t) - f_t(x^*)\right] \to 0$ and thus convergence. First, we shall say something about each term $\langle \tilde{g}_t, \tilde{x}_t - x^* \rangle$.

**Lemma 2:**  If $X = \mathbb{R}^n$, then for all $x^*$,

$$\langle \tilde{x}_t - x^*, \tilde{g}_t \rangle = \frac{1}{2}\eta_t \|\tilde{g}_t\|^2 + \frac{D\left(x^* \| x_t\right) - D\left(x^* \| x_{t+1}\right)}{\eta_t} + \left[ \sum_{i \in A_t} \eta_i \langle \tilde{g}_i, \tilde{g}_t \rangle - \sum_{i \in B_t} \eta_i \langle \tilde{g}_i, \tilde{g}_t \rangle \right]$$

**Proof:**

$$
\begin{aligned}
D\left(x^* \| x_{t+1}\right) - D\left(x^* \| x_t\right) &= \frac{1}{2}\|x^* - x_t + x_t - x_{t+1}\|^2 - \frac{1}{2}\|x^* - x_t\|^2 \\
&= \frac{1}{2}\|x^* - x_t + \eta_t \tilde{g}_t\|^2 - \frac{1}{2}\|x^* - x_t\|^2 \\
&= \frac{1}{2}\eta_t^2 \|\tilde{g}_t\|^2 - \eta_t \langle x_t - x^*, \tilde{g}_t \rangle \\
&= \frac{1}{2}\eta_t^2 \|\tilde{g}_t\|^2 - \eta_t \langle \tilde{x}_t - x^*, \tilde{g}_t \rangle - \eta_t \langle x_t - \tilde{x}_t, \tilde{g}_t \rangle
\end{aligned}
$$

Expand the last term:

$$\langle x_t - \tilde{x}_t, \tilde{g}_t \rangle = \left\langle \left[ -\sum_{i \in A_t} \eta_i \tilde{g}_i + \sum_{i \in B_t} \eta_i \tilde{g}_i \right], \tilde{g}_t \right\rangle$$

$$= -\sum_{i \in A_t} \eta_i \langle \tilde{g}_i, \tilde{g}_t \rangle + \sum_{i \in B_t} \eta_i \langle \tilde{g}_i, \tilde{g}_t \rangle$$

Therefore

$$D\left(x^* \| x_{t+1}\right) - D\left(x^* \| x_t\right) = \frac{1}{2}\eta_t^2 \|\tilde{g}_t\|^2 - \eta_t \langle \tilde{x}_t - x^*, \tilde{g}_t \rangle - \eta_t \left[ -\sum_{i \in A_t} \eta_i \langle \tilde{g}_i, \tilde{g}_t \rangle + \sum_{i \in B_t} \eta_i \langle \tilde{g}_i, \tilde{g}_t \rangle \right]$$

$$\frac{D\left(x^* \| x_{t+1}\right) - D\left(x^* \| x_t\right)}{\eta_t} = \frac{1}{2}\eta_t \|\tilde{g}_t\|^2 - \langle \tilde{x}_t - x^*, \tilde{g}_t \rangle + \left[ \sum_{i \in A_t} \eta_i \langle \tilde{g}_i, \tilde{g}_t \rangle - \sum_{i \in B_t} \eta_i \langle \tilde{g}_i, \tilde{g}_t \rangle \right]$$

$$\langle \tilde{x}_t - x^*, \tilde{g}_t \rangle = \frac{1}{2}\eta_t \|\tilde{g}_t\|^2 + \frac{D\left(x^* \| x_t\right) - D\left(x^* \| x_{t+1}\right)}{\eta_t} + \left[ \sum_{i \in A_t} \eta_i \langle \tilde{g}_i, \tilde{g}_t \rangle - \sum_{i \in B_t} \eta_i \langle \tilde{g}_i, \tilde{g}_t \rangle \right].$$

This completes the proof of Lemma 2. □

**Back to Theorem 1:**  Returning to the proof of Theorem 1, we use Lemma 2 to expand the regret $R[X]$:

$$R[X] \le \sum_{t=1}^{T} \langle \tilde{g}_t, \tilde{x}_t - x^* \rangle = \sum_{t=1}^{T} \frac{1}{2}\eta_t \|\tilde{g}_t\|^2 + \sum_{t=1}^{T} \frac{D\left(x^* \| x_t\right) - D\left(x^* \| x_{t+1}\right)}{\eta_t}$$

$$+ \sum_{t=1}^{T} \left[ \sum_{i \in A_t} \eta_i \langle \tilde{g}_i, \tilde{g}_t \rangle - \sum_{i \in B_t} \eta_i \langle \tilde{g}_i, \tilde{g}_t \rangle \right]$$

$$= \sum_{t=1}^{T} \left[ \frac{1}{2}\eta_t \|\tilde{g}_t\|^2 + \sum_{i \in A_t} \eta_i \langle \tilde{g}_i, \tilde{g}_t \rangle - \sum_{i \in B_t} \eta_i \langle \tilde{g}_i, \tilde{g}_t \rangle \right]$$

$$+ \frac{D\left(x^* \| x_1\right)}{\eta_1} - \frac{D\left(x^* \| x_{T+1}\right)}{\eta_T} + \sum_{t=2}^{T} \left[ D\left(x^* \| x_t\right) \left( \frac{1}{\eta_t} - \frac{1}{\eta_{t-1}} \right) \right]$$

We now upper-bound each of the terms:

$$\sum_{t=1}^{T} \frac{1}{2}\eta_t \|\tilde{g}_t\|^2 \le \sum_{t=1}^{T} \frac{1}{2}\eta_t L^2 \qquad \text{(Lipschitz assumption)}$$

$$= \sum_{t=1}^{T} \frac{1}{2} \frac{\sigma}{\sqrt{t}} L^2$$

$$\le \sigma L^2 \sqrt{T},$$

and

$$\frac{D\left(x^* \| x_1\right)}{\eta_1} - \frac{D\left(x^* \| x_{T+1}\right)}{\eta_T} + \sum_{t=2}^{T} \left[ D\left(x^* \| x_t\right) \left( \frac{1}{\eta_t} - \frac{1}{\eta_{t-1}} \right) \right]$$

$$\le \frac{F^2}{\sigma} + 0 + \frac{F^2}{\sigma} \sum_{t=2}^{T} \left[ \sqrt{t} - \sqrt{t-1} \right] \qquad \text{(Bounded diameter)}$$

$$= \frac{F^2}{\sigma} + \frac{F^2}{\sigma} \left[ \sqrt{T} - 1 \right]$$

$$= \frac{F^2}{\sigma} \sqrt{T},$$

and

$$\sum_{t=1}^{T} \left[ \sum_{i \in \mathcal{A}_t} \eta_i \langle \tilde{g}_i, \tilde{g}_t \rangle - \sum_{i \in \mathcal{B}_t} \eta_i \langle \tilde{g}_i, \tilde{g}_t \rangle \right]$$

$$\leq \sum_{t=1}^{T} \left[ |\mathcal{A}_t| + |\mathcal{B}_t| \right] \eta_{\max(1, t-(s+1)P)} L^2 \qquad \text{(from Lemma 1: } \min(\mathcal{A}_t \cup \mathcal{B}_t) \geq \max(1, t - (s+1)P))$$

$$= L^2 \left[ \sum_{t=1}^{(s+1)P} \left[ |\mathcal{A}_t| + |\mathcal{B}_t| \right] \eta_1 + \sum_{t=(s+1)P+1}^{T} \left[ |\mathcal{A}_t| + |\mathcal{B}_t| \right] \eta_{t-(s+1)P} \right] \qquad \text{(split the sum)}$$

$$= L^2 \left[ \sum_{t=1}^{(s+1)P} \left[ |\mathcal{A}_t| + |\mathcal{B}_t| \right] \sigma + \sum_{t=(s+1)P+1}^{T} \left[ |\mathcal{A}_t| + |\mathcal{B}_t| \right] \frac{\sigma}{\sqrt{t-(s+1)P}} \right]$$

$$\leq \sigma L^2 \left[ \sum_{t=1}^{(s+1)P} 2s(P-1) + \sum_{t=(s+1)P+1}^{T} 2s(P-1) \frac{1}{\sqrt{t-(s+1)P}} \right] \qquad \text{(from Lemma 1: } |\mathcal{A}_t| + |\mathcal{B}_t| \leq 2s(P-1))$$

$$\leq 2\sigma L^2 s(P-1) \left[ (s+1)P + 2\sqrt{T-(s+1)P} \right] \qquad \left( \text{Note that } \sum_{i=a}^{b} \frac{1}{2\sqrt{i}} \leq \sqrt{b-a+1} \right)$$

$$\leq 2\sigma L^2 s(P-1) \left[ (s+1)P + 2\sqrt{T} \right]$$

$$\leq 2\sigma L^2 \left[ (s+1)P \right]^2 + 4\sigma L^2 (s+1)P\sqrt{T}.$$

Hence,

$$R[X] \leq \sum_{t=1}^{T} \langle \tilde{g}_t, \tilde{x}_t - x^* \rangle \quad \leq \quad \sigma L^2 \sqrt{T} + F^2 \frac{\sqrt{T}}{\sigma} + 2\sigma L^2 \left[ (s+1)P \right]^2 + 4\sigma L^2 (s+1)P\sqrt{T}.$$

If we set the initial step size $\sigma = \frac{F}{L\sqrt{2\kappa}}$ where $\kappa = (s+1)P$, then

$$R[X] \quad \leq \quad \frac{FL\sqrt{T}}{\sqrt{2\kappa}} + FL\sqrt{2\kappa T} + \sqrt{2}FL\kappa^{3/2} + 2FL\sqrt{2\kappa T}$$

$$= \quad FL\sqrt{2\kappa T} \left[ 3 + \frac{1}{2\kappa} + \frac{\kappa}{\sqrt{T}} \right].$$

Assuming $T$ large enough that $\frac{1}{2\kappa} + \frac{\kappa}{\sqrt{T}} \leq 1$, we get

$$R[X] \quad \leq \quad 4FL\sqrt{2\kappa T}.$$

This completes the proof of Theorem 1. $\square$

In Langford *et al.* the error between $x_t$ and $\tilde{x}_t$ consists of exactly $\tau$ terms, where $\tau$ is the fixed-delay parameter of their system. In contrast, the same error under SSP contains at most $2(s+1)P$ terms, meaning that the actual convergence rate can be improved (by up to a constant factor) with a good SSP implementation. We also note that Theorem 1 does not address the other key feature of SSPtable, namely that workers spend less time waiting for the network, due to caching. In practice, while increasing the staleness of SSPtable decreases the per-iteration convergence rate (as Theorem 1 suggests), it also increases the number of iterations executed per unit time. The result is *faster* convergence with increased staleness, up to a point.