[Reviews · NeurIPS 2013]

Submitted by Assigned_Reviewer_2

The paper describes a bounded-staleness design for a parameter server for iterative distributed learning based on vector clocks. The idea is clear, intuitive and provides a nice parameterized range between the previous designed for pure asynchronous updates (e.g., Hogwild or Yahoo LDA) and approaches that have a hard synchronization barrier (e.g., most recent SGD papers). While there are no breaking theoretical insights, the paper correctly adapts the analysis from [17]. THe paper is very well-written, and the SSPtable design description is exemplary.

The experimental results are convincing, with computation-vs-communication and clocks/worker/time results particularly encouraging . A few areas for improvement/details that would help:
- Lasso results are painfully limited to a toy synthetic dataset -- a more comprehensive evaluation on it would be quite useful.
- A direct comparison with state-of-the-art distributed learning packages (Yahoo LDA, Hogwild) would be illustrative.
- There should be a single-worker baseline.
- How exactly is data partitioned? This should be mentioned.

A couple suggestions:
- For matrix factorization, could the idea be combined with Gemulla et al's approach for sub-epochs that operate on non-overlapping parameter blocks sequentially?
- It would be helpful to discuss the possibility of forcing the slowest stragglers to sync with an incomplete iteration if that prevents blocking for current staleness level.
Summary: The paper is well-written, describes an intuitive idea, and provides convincing experimental results with some relatively minor areas for improvement.

Submitted by Assigned_Reviewer_7

The paper presents an approach to building a parameter server for distribute ML systems that presents a view to each client where parameters have a bounded degree of staleness. Using a combination of caches, the client interface guarantees that all updates to the parameter array occurring after a fixed deadline (the current clock/iteration/tick minus a fixed delay) are visible along with more recent updates if possible. Thus the interface presents a view of parameters that integrates most updates along with best-effort service for more recent updates. It is shown that this simple semantic preserves the theoretical guarantees of cyclic delay methods while being significantly faster in practice. Empirical analysis on several problems with multiple cluster configurations show that the advantage is due to a combination of increased efficiency (over BSP) and optimization progress per update (over Asynchronous).

This paper presents a simple set of semantics that improve on "parameter server" schemes currently showing up in large scale machine learning applications. Along with straight-forwardly carrying over simple theoretical guarantees, the method is apparently significantly faster for reasonably-sized tests than the obvious competitors and on those points alone I think this is pretty positive.

There are a lot of well-known problems in loaded clusters (stragglers, etc.) and as far as I can tell this approach should deal with them well. A saving grace may be that the caching mechanism reduces overhead dramatically for the slowest nodes thus giving a degree of "load balancing" that is annoying hard to get in other ways. Is it possible to show that this is actually happens to a degree sufficient to enable the "catch up" phenomenon claimed in the paper?

Some discussion of the read semantics is given with details on the cache policy (which just falls through when the local cache is stale). Due to the read-my-writes policy, is it correct that all local writes are written through to all caches/server? A sentence of clarification on the writing semantics might help for reproduction of the work.

As far as systems papers go the authors make a good case for the approach.

Pros:
Fairly simple semantics.
Preserves basic theoretical guarantees for typical methods (e.g., SGD).
Experimental analysis shows actual speedups on multiple distributed problems and the speedup comes from the sources expected.

Cons:
A comparison to cyclic delay would be nice.
Potentially complex to re-implement in comparison to asynchronous/BSP; is it worth it for the moderate speedup?

**Update
Thanks to the authors for their clarifications and additions. I think these are reasonable arguments and resolve my [very limited] qualms.
Summary: A framework for parameter servers in machine learning applications is presented with moderate speedups and appealing theoretical and practical properties.

Submitted by Assigned_Reviewer_8

The paper proposes Stale Synchronous Parallel(SSP) computation for machine learning. It also discusses an implementation thereof and, most importantly, provides the first discussion of its relationship to machine learning quantities. The latter analysis is done for SGD. Comprehensive experiments show the empirical validity of the approach.

The paper is very well written and easy to follow. It has been a pleasure to read and review. Thanks for that!

While my overall verdict is to accept the paper, I'd like point out a few things that might improve the clarity of exposition and the embedding into the related work. To my mind, the main contribution of the work is not to _suggest_ to use SSP for machine learning. That (arguably somewhat obvious) connection has been made before, if only amongst practitioners in the field and not in the academic discourse.

The key contribution in my mind is to (a) have done it with rigor and (b), more importantly, to provide analysis that connects systems level parameters like stale-ness with machine learning concepts such as convergence of the learner. If at all possible, more space should be devoted to that analysis, as there is a suggestion in the paper that space was a consideration when leaving examples beyond SGD out. Please add them to the paper or to the appendix.

Lastly, some nitpicking: The statement that SSPtable is more general than Hadoop on page 7, Line 376 is a bit naive in the following sense: SSPtable is optimized for small incremental updates, while Hadoop can deal with massive objects being aggregated. Also, Hadoop has a (rather good) shuffle phase, while SSPtable doesn't need one. I'd thus suggest to weaken that statement a bit as it does stand out a bit in an otherwise clean paper.
Summary: Excellent paper that makes the connection between SSP and machine learning. The presentation and content are great, an easy accept.
Author Feedback

Author rebuttal: We thank the reviewers for their quality feedback, and their encouraging assessment of our paper. Below, please find our responses to each reviewer:

R2

On data partitioning: our implementation statically partitions data equally over worker threads, for expedience. For example, with N data points and P total threads, every thread gets N/P data points. However, SSP does not require static partitioning, and we want to explore dynamic load balancing with staleness in future work. For LDA, a data point is one document. For MF, a data point is one row of the input matrix. Lasso is an exception: being a coordinate-parallel algorithm, we partition evenly over input dimensions instead of samples. We will put this information in the appendix.

On Lasso evaluation, we used the commonly-adopted coordinate-parallel Shotgun algorithm, which has a parallelism limit that depends on correlation in the input data. This limited the number of dimensions (which in our opinion is more interesting than sample size) in our experiments, for BSP Shotgun would not converge with larger datasets or more machines (and, not unexpectedly, neither did SSP). Thus, our reported results are the largest setting on which Shotgun converged reliably.

We will include single-worker baselines: going from 1 to 32 machines gives a ~10x convergence speedup. On YahooLDA and Hogwild, note that we have compared SSP with async updates, which those packages are based upon. We are also planning a direct comparison as future work, the main issue being inference algorithm differences: for example, YahooLDA uses the Yao et al. (KDD 2009) fast LDA sampler, whereas we intentionally chose simple, exemplar algorithms to demonstrate that SSP is widely applicable --- for LDA, we chose the common Griffiths and Steyvers (PNAS 2004) collapsed Gibbs sampler. To ensure a fair comparison, we plan to reimplement YahooLDA and Hogwild's chosen algorithms on SSP.

Combining SSP with Gemulla et al.'s strategy and syncing on incomplete iterations are indeed interesting ideas. Note that the latter might require programmers to explicitly define when syncing is allowed.

R7

We've observed that SSP indeed produces "catch up", but the amount depends on a variety of factors. A full analysis of these factors as a function of system, algorithm and data characteristics is an area for future study.

On read-my-writes, we use a write-back policy: all writes immediately go to the thread cache; writes are flushed to the process cache upon clock(), and flushed to the servers after all threads have called clock(). We will clarify this in the paper.

On design complexity, our async and BSP implementations enjoy many of the same optimizations as SSP, which enables fair comparisons between all three. Thus, all three implementations are of nearly comparable complexity, while straightforward implementations of async and BSP would have been considerably slower. Moreover, we view SSPTable as part of a greater system for large-scale ML, with the complexity laying the groundwork for new extensions. For example, SSPTable could be extended to allow different staleness on each read, supporting algorithms that exploit varying staleness on different parts of the model, or task schedulers that dynamically reassign data and work to different machines. We think these are worthy goals to pursue, in addition to pure performance.

While the cyclic delay system is theoretically close to SSP, it has less in common at the systems level with SSP, than SSP does with async and BSP. Briefly, this is because cyclic delay puts an ordering on workers, unlike BSP, SSP and async. Our comparison between BSP, SSP and async maintains fairness by enforcing similar implementation complexites, via reuse of ML algorithm, networking and cache code. This level of reuse is not possible with cyclic delay (due to the ordering issue), and we feel a fair comparison with cyclic delay is best left as future work.

R8

We appreciate the reviewer’s statement that our paper provides the first rigorous analysis of SSP behavior in practical ML algorithms, connecting systems level parameters like staleness with ML concepts like convergence. It is true that bounded staleness updates have been practiced in the industrial setting, as a necessary technique (albeit sans justification) in adapting to real distributed environments. Sometimes, bounded staleness might be found to empirically improve speed, though other times, it might compromise convergence quality and hence go unreported. To our knowledge, the convergence speed and quality effects of bounded staleness practices have not yet been studied in a rigorous and systematic manner that covers a spectrum of ML algorithms under different conditions, with both theoretical guarantees and empirical backing. As with the reviewer, we believe such a principled view is important, because it allows practitioners and the academic community to exploit bounded staleness more easily and correctly - leading to more ML problems benefiting from SSP. We will state our contributions more accurately in the final version.

We appreciate the reviewer’s interest in more theoretical analysis beyond SGD. One very promising avenue concerns distributed block-gradient algorithms for problems like Lasso, under both data parallelization (in which samples are distributed, and which SSPtable supports), as well as model parallelization (in which smaller parts of a big model are distributed). Such analysis is currently preliminary, as it not only involves insights that go beyond SSP (and hence is beyond the scope of this paper), but is also inherently difficult due to model decomposition and data partitioning issues being intertwined. We will clarify this in the final version.

Our SSP vs. Hadoop claim was certainly too broad, and only focused on the BSP computational model in Hadoop. We will adjust the statement to more precisely reflect the pros and cons of each method.